# Clinical Factors Contributing to Cognitive Function in the Acute Stage after Treatment of Intracranial Aneurysms: A Cross-Sectional Study

**DOI:** 10.3390/jcm11175053

**Published:** 2022-08-28

**Authors:** Yeo Jin Kim, Sang-Hwa Lee, Jin Pyeong Jeon, Hui-Chul Choi, Hyuk Jai Choi

**Affiliations:** 1Department of Neurology, Chuncheon Sacred Heart Hospital, Hallym University College of Medicine, Chuncheon-si 24253, Korea; 2Department of Neurosurgery, Chuncheon Sacred Heart Hospital, Hallym University College of Medicine, Chuncheon-si 24253, Korea

**Keywords:** cognitive changes, intracranial aneurysm, treatment, subarachnoid haemorrhage, endovascular coiling, microsurgical clipping

## Abstract

Background: The factors affecting cognitive function after treatment of subarachnoid haemorrhage (SAH) can be categorised into aneurysmal factors, procedural factors, and complications. The aim of this study was to investigate which of these factors has greater influence on the cognitive function. Methods: We retrospectively identified 14 patients with unruptured intracranial aneurysms (UIAs) and 34 patients with SAH with mild symptoms at disease onset (Hunt and Hess grade: >3). All patients underwent neuropsychological tests within 35 days of discharge from hospitalisation for treatment. The relationship between the clinical factors and each neuropsychological test score was evaluated using multiple linear regression analysis after controlling for age and years of education. Results: Patients with UIA showed greater cognitive impairment in visual memory and the frontal/executive domains. Hypertension was associated with cognitive impairment. Patients with SAH showed greater cognitive impairment in the visuospatial, verbal memory, and frontal/executive domains. The dome-to-neck ratio, aneurysms located in the posterior circulation, microsurgical clipping, procedure time, anaesthesia duration, and complications were associated with cognitive impairment. Conclusions: Underlying diseases, procedural factors, and complications contributed to cognitive impairment after treatment of intracranial aneurysms. Since the effect of each factor on each cognitive domain was slightly different, a more in-depth study of these effects is needed.

## 1. Introduction

Although the prognosis of spontaneous subarachnoid haemorrhage (SAH) has improved significantly compared with the past due to the advances in treatment technology, many patients still suffer from sequelae including cognitive impairment [1]. In particular, patients who do not show cognitive impairment on a simple general cognitive assessment often complain of cognitive impairment. Since 80% of the cases of SAH are caused by aneurysm rupture [2], in order to reduce the sequelae of SAH, doctors attempt to detect and treat intracranial aneurysms before they rupture. However, even the treatment of an unruptured intracranial aneurysm (UIA) is known to affect the cognitive function, although the mechanisms underlying this are not well understood [3]. A previous study reported post-treatment cognitive impairment in 5.5% of the patients with UIA but without a history of SAH [4]. However, the clinical factors responsible for cognitive impairment after treatment of both UIA and SAH remain unclear.

Patients with a ruptured anterior communicating artery (AcomA) aneurysm have exhibited impaired memory and executive functions with behaviour deficit [5,6]. One study reported an inhibition failure [7] and another demonstrated decreases in the motor speed and perceptual vigilance [8] in patients with ruptured AcomA aneurysms. These patients also reportedly exhibit apathy or a lack of energy [9]. Therefore, the medial frontal structures are believed to be involved in patients with ruptured AcomA aneurysms [10]. Relatively few studies have been conducted on patients with ruptured aneurysms at sites other than the AcomA. One study reported that patients with ruptured middle cerebral artery (MCA) aneurysms showed impaired verbal and visual short-term memory [11].

A domain-specific approach, rather than general assessment, could help evaluate the cognitive function more accurately. Since different cognitive domains may be affected depending on the characteristics of the disease, a general cognitive assessment may confound the findings. In one study, the results of general cognitive assessment and domain-specific cognitive function measurement were contradictory [12]. Therefore, it is preferable to perform a domain-specific cognitive function measurement; however, the number of studies on this is relatively small. As a result, the sensitive evaluation of cognitive impairment cannot be performed satisfactorily [13].

The characteristics of the acute stage of brain damage are different from those of the chronic stage of brain damage, which is due to the fact that the brain has the plasticity to repair the damage [14]. Therefore, the direct effects of a brain injury are more pronounced in the acute phase [14]. Previous studies have noted differences in the cognitive function between the acute stage and a long-term follow-up stage. In some studies, the cognitive function at long-term follow-up was noted to have recovered significantly from its status at the acute stage follow-up [12,15]; however, many existing studies have only investigated the cognitive function after 3 months, i.e., after the acute stage. In most cases, the assessment comprised a general cognitive assessment [1,16,17,18]. Therefore, the domain-specific cognitive function during the acute phase after the treatment remains unclear.

Therefore, in our study, we investigated the post-treatment cognitive function of patients with intracranial aneurysms and investigated the clinical factors that affected the cognitive function in each domain in the acute stage.

## 2. Materials and Methods

### 2.1. Participants

We consecutively and retrospectively identified participants from an observational database of patients who were treated for intracranial aneurysms at the cerebrovascular accident centre of the Chuncheon Sacred Heart Hospital between March 2014 and December 2017. Forty-eight patients were recruited according to the following criteria: (1) One or more intracranial aneurysms detected on an angiogram and CT scan, (2) a subjective cognitive complaint by the patient or his/her caregiver, and (3) neuropsychological testing performed within 35 days of discharge from hospitalisation for the microsurgical or endovascular treatment. Patients with SAH were assessed using the Hunt and Hess scale score [19] at admission; the scores ranged from 1 to 3. Furthermore, the patients did not have localised neurological deficit, including cortical signs (such as neglect, apraxia or aphasia). All patients underwent microsurgical clipping or endovascular coiling, and no parenchymal damage was observed on post-treatment imaging. Immediately after the treatment, the absence of gross haemorrhage and low density was confirmed on computed tomography. Magnetic resonance images, including diffusion-weighted images, apparent diffusion coefficient maps, gradient echo sequences, and magnetic resonance angiography images, were acquired within 48 h of the treatment.

The detailed demographic and clinical characteristics of the participants are presented in Table 1. Among the 48 patients with intracranial aneurysms that were identified, 14 had UIA and 34 had experienced SAH. The average age of the patients with UIA was higher than the patients with SAH (*p* = 0.003). The proportion of patients with hypertension was also higher among those with UIA than among those with SAH (*p* = 0.015). Compared with patients with SAH, a higher proportion of patients with UIA underwent microsurgical clipping (*p* = 0.019). The proportion of patients with a new infarction was also lower among those with UIA than among those with SAH (*p* = 0.030).

### 2.2. Standard Protocol Approvals, Registrations, and Patient Consent

This study was approved by the institutional review board of the Chuncheon Sacred Heart Hospital (CHUNCHEON 2021-06-018). Due to the retrospective nature of the study, the institutional review board approved data collection without informed consent prior to the study.

### 2.3. Distribution of Intracranial Aneurysms

Table 2 shows the distribution of the intracranial aneurysms according to the location. In particular, 12 of the 14 patients with UIA had a single aneurysm. Additionally, the aneurysms were located in the internal carotid artery, AcomA, MCA, and posterior communicating artery (PcomA) in 2, 2, 5, and 3 of these 14 patients, respectively. Moreover, 30 of the 34 patients with SAH had a single aneurysm. Furthermore, the aneurysms were located in the internal carotid artery, AcomA, anterior cerebral artery (ACA), MCA, PcomA, posterior inferior cerebellar artery (PICA), and vertebral artery (VA) in 2, 12, 1, 6, 7, 1, and 1 of these 34 patients, respectively.

### 2.4. Procedures

The patients were operated on by two neurosurgeons (HJC and JPJ). The neurosurgeons decided on the appropriate treatment based on the following factors: (1) The patient’s age and overall medical condition and (2) the aneurysm’s location, size, morphological features, and relationship with the adjacent vessels. Among the 14 patients with UIA, 10, 3, and 1 underwent microvascular clipping, endovascular coiling, and both, respectively. Among the 34 patients with SAH, 11, 20, and 3 underwent microvascular clipping, endovascular coiling, and both, respectively. For the patients who underwent microvascular clipping, the standard neurosurgical clipping method was used. This involved an open craniotomy with a pterional approach and dissection of the vessels of the circle of Willis within the basal cisterns via the Sylvian fissure. After the aneurysm was dissected free from the surrounding brain and vessels, the clips were placed across the neck of the aneurysm. After clipping, intraoperative doppler and indocyanine green angiography were performed to check the blood flow. Among the patients who underwent clipping, 1 patient with SAH exhibited an incomplete occlusion; therefore, additional coiling was performed for the remaining portion. In patients who underwent endovascular coiling, a guiding catheter was placed in the internal carotid artery or the vertebral artery, and the aneurysm was probed using a microcatheter (Prowler 14, Codman & Shurtleff, Inc. Raynham, MA, USA) and a microwire (Synchro-14, Boston Scientific Corporation, Natick, MA, USA). Endovascular selective occlusion of the aneurysm sac with preservation of the parent artery was performed. Among the 27 patients who underwent endovascular coiling, 8 patients received stent-assisted coil embolisation. Moreover, 20 and 3 of the 27 patients received anticoagulation therapy with intravenous heparin and antiplatelet loading, respectively.

### 2.5. Neuropsychological Tests

All patients underwent neuropsychological tests comprising the Seoul Neuropsychological Screening Battery. This battery includes quantitative tests, including the digit span test (forward and backward), Korean version of the Boston Naming Test (K-BNT), Seoul Verbal Learning Test (SVLT; three learning-free recall trials of 12 words, a 20 min delayed recall trial for these 12 items, and a recognition test), Rey-Osterrieth Complex Figure Test (copying, immediate, 20 min delayed recall, and recognition), semantic and phonemic Controlled Oral Word Association Test (COWAT), and Stroop Test (word and colour reading of 112 items during a 2 min period). The digit span test was used to evaluate attention, while the K-BNT was used to evaluate the language function. The immediate recall, delayed recall, and recognition aspects of the SVLT and the RCFT were used to evaluate the verbal and visual memory functions, respectively. The RCFT copy score was used to evaluate the visuospatial functions. The COWAT and the Stroop test were used to evaluate the frontal/executive functions. For comparison, we used raw scores and standardised scores. The raw scores were transformed into standardised scores (z score) based on the means of normal data from 1067 individuals (http://www.humanbrainkorea.com/Item/Default.aspx?sub=SNSB_2 (accessed on 1 June 2021)), which are determined by differences in the age and educational level. The scores are considered to be abnormal when they are lower than the −1.0 standard deviation (16th percentile) of the age- and education-adjusted norms. This test is applicable to people between the ages of 45 and 90 years. The tests were administered and scored by a trained neuropsychologist in a dedicated test room. The entire test took approximately 1.5–2 h to complete.

### 2.6. Statistical Analyses

The relationships between the clinical factors and each neuropsychological test score were evaluated using multiple linear regression analysis; the clinical factors were considered as the determinants, while the neuropsychological test scores were considered as the outcome variables after controlling for the age and years of education. If multiple factors influenced the neuropsychological test results, we performed further analysis using multiple linear regression with all the clinical factors that were significantly associated with the cognitive function in the former analysis (*p* < 0.05). A two-tailed *p* value of <0.05 was deemed significant. Statistical analysis was performed with PASW Statistics 20 (SPSS, Chicago, IL, USA). To evaluate the mediation effects of the clinical factors on the cognitive function, path analyses were performed after controlling for the age and years of education. The Amos Version 18.0 software (SPSS, Chicago, IL, USA) was used for all path analyses performed with maximum likelihood estimation.

## 3. Results

### 3.1. Neuropsychiatric Features after Treatment of Intracranial Aneurysms

Patients with UIA performed 1.02, 1.29, and 1.60 standard deviations (SDs) below the published norms on the RCFT immediate recall, COWAT animal items, and Stroop test colour reading, respectively. Patients with SAH performed 1.10, 1.12, 1.45, 1.18, 1.23, and 1.83 SDs below the published norms on the RCFT copy score, SVLT immediate recall, SVLT delayed recall, COWAT animal items, COWAT phonemic items, and Stroop test colour reading, respectively (Table 3).

### 3.2. Factors Contributing to Cognitive Impairment after Treatment of Intracranial Aneurysms

In patients with UIA, among the clinical factors, hypertension was associated with an impaired RCFT immediate recall (Table 4). In patients with SAH, among the clinical factors, microsurgical clipping, procedure time, and anaesthesia duration were associated with an impaired RCFT copy. Hydrocephalus was associated with impaired SVLT immediate and delayed recall. The dome-to-neck ratio of the aneurysm, microsurgical clipping, vasospasm, and new infarction were associated with COWAT animal items. The procedure time and the anaesthesia duration were associated with impaired COWAT phonemic items. Distribution of the aneurysms was associated with an impaired Stroop test colour reading (Table 5).

There were several factors that affected the RCFT copy, COWAT animal item, and COWAT phonemic item scores. Therefore, the factors affecting each test were analysed in a single model. However, none of the tested factors had any significant influence.

### 3.3. Mediation of Clinical Factors on Cognitive Function (Figure 1)

We performed a path analysis using factors significantly correlated with the RCFT copy score, i.e., microsurgical clipping, procedure time, and anaesthesia duration. These results showed good to fit to the data: Chi-square = 1.561, degree of freedom = 2, *p* = 0.458, comparative fit index = 1.000, and root mean square error of approximation <0.0001. Microsurgical clipping, prolonged procedure time, and prolonged anaesthetic duration were associated with a decreased RCFT copy score. The prolonged procedure time was also associated with the anaesthesia duration; however, the anaesthetic duration itself was not associated with the RCFT copy score.

**Figure 1 jcm-11-05053-f001:**
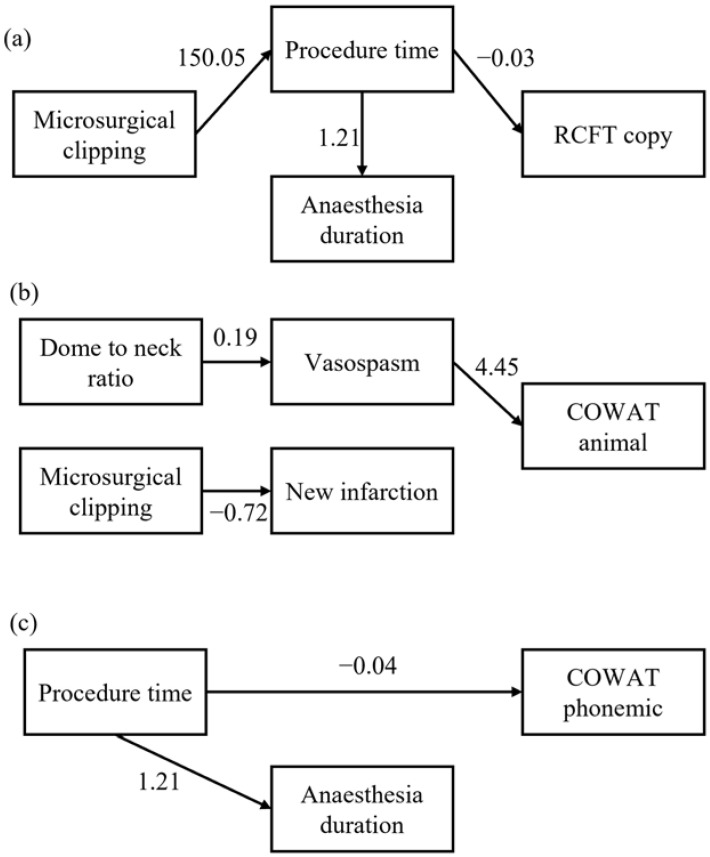
Schematic diagram of the path analyses for cognitive function. (**a**) Microsurgical clipping was considered as a predictor. Procedure time was considered as a mediator variable for the Rey-Osterrieth complex figure test copy score. Age and years of education were considered as the covariates. (**b**) The dome-to-neck ratio and microsurgical clipping were considered as predictors. Vasospasm was considered as a mediator variable for the controlled oral word association test animal item score. Age and years of education were considered as the covariates. (**c**) Procedure time was considered as a predictor for the controlled oral word association test phonemic item score. Age and years of education were considered as the covariates.

Furthermore, we performed a path analysis using factors significantly correlated with the COWAT animal item scores, namely the dome-to-neck ratio, microsurgical clipping, vasospasm, and new infarction. These results showed good to fit to the data: Chi-square = 1.213, degree of freedom = 1, *p* = 0.271, comparative fit index = 0.996, and root mean square error of approximation = 0.080. The dome-to-neck ratio was associated with vasospasm, while vasospasm was associated with an increased COWAT animal item score. Microsurgical clipping was associated with new infarction.

Finally, we performed a path analysis using factors that were significantly correlated with the COWAT phonemic item scores, namely the procedure time and anaesthesia duration. These results showed good to fit to the data: Chi-square = 1.418, degree of freedom = 1, *p* = 0.234, comparative fit index = 0.997, and root mean square error of approximation = 0.122. Prolonged procedure time was associated with a decreased COWAT phonemic item score. It was also associated with the anaesthesia duration; however, the anaesthesia duration was not associated with the COWAT phonemic item score.

## 4. Discussion

Patients with intracranial aneurysms exhibited a cognitive impairment in the neuropsychological tests performed in the acute stage of treatment. Patients with UIA exhibited a lower memory and frontal/executive functions than the published norm. Patients with SAH showed lower visuospatial function, memory, and frontal/executive functions than the published norm. In patients with UIA, a history of hypertension was associated with an impaired cognitive function. In patients with SAH, the dome-to-neck ratio, aneurysms located in the posterior circulation, treatment with microsurgical clipping, procedure time, anaesthesia duration, vasospasms, and new infarctions were associated with an impaired cognitive function.

In the cognitive function tests performed in the acute stage of treatment, patients with intracranial aneurysms exhibited lower visuospatial, memory, and frontal/executive functions than the published norms. Previous studies have consistently reported cognitive impairments in patients with ruptured intracranial aneurysms [20]. The memory and frontal/executive functions were the most frequently impaired cognitive functions in patients with vascular damage [21,22,23]; however, for patients with UIA, previous studies have reported controversial results. Some studies have reported that patients with UIAs did not show a cognitive impairment, [24,25,26], whereas others have demonstrated a cognitive impairment after treatment of UIAs [4,17]. A previous study showed that patients with both ruptured and unruptured aneurysms exhibited a decreased cognitive function for word fluency, verbal recall, and frontal lobe function, similar to the findings in the present study [17].

Cognitive impairment after treatment of intracranial aneurysms could be due to more diffuse and indirect effects [27]. Some studies have reported that aneurysm rupture in a specific location, such as the AcomA, causes impairment of specific cognitive functions, such as the memory and frontal/executive functions. However, other studies have reported that the location of the involved vessels is not significantly related to a cognitive impairment [17,28]. A previous functional magnetic resonance imaging (fMRI) study on patients treated for ruptured intracranial aneurysms revealed that an abnormal activity, mainly in the parahippocampal gyrus, left inferior temporal gyrus, and left thalamus, was positively correlated with memory performance [29]. Another fMRI network study on patients with ruptured intracranial aneurysms also revealed that a frontal-parietal control network (comprising the dorsolateral prefrontal cortex, medial frontal gyri, superior frontal gyri, middle frontal gyri, precuneus, and inferior parietal cortex) was linked to a cognitive deficit; the underlying mechanisms, however, remained unclear [30]. In a magnetoencephalography study, increased neuronal activity in the absence of structural lesions was noted in the posterior and anterior cingulate gyri in patients with ruptured aneurysms during visual working memory tasks [31]; however, there has been no functional imaging study on patients with UIAs. Therefore, how functional changes in the brain result in cognitive deficits in patients with UIAs remain unknown.

In this study, the treatment with microsurgical clipping was associated with poor visuospatial and frontal/executive functions. Previous studies have reported that cognitive impairment was more prominent in patients who underwent microsurgical clipping than in those who underwent endovascular coiling [18]. A randomised clinical trial on patients with ruptured intracranial aneurysms reported a higher mortality and morbidity for microsurgical clipping than for endovascular coiling [32]. Therefore, compared with endovascular coiling, microsurgical clipping seems to have a greater impact on the cognitive function [33]. This is due to the fact that microsurgical clipping can cause a direct mechanical injury to the brain through frontal lobe retraction [17]. Microsurgical clipping also causes decreased cerebral perfusion and brain damage, which could impair the cognitive function [34]. Particularly in this study, microsurgical clipping did not affect the cognitive function in the UIA group; however, it affected the cognitive function in the SAH group. This might indicate that the aneurysm rupture and effects of microsurgical clipping combined to produce a poor prognosis. Although microsurgical clipping does not affect the brain prior to haemorrhagic injury, in the case of brain damage due to haemorrhage, microsurgical clipping may have a secondary effect on cognitive impairment without noticeable brain injury.

Additionally, in this study, a long procedure time was associated with poor visuospatial and frontal/executive functions. Patients treated with microsurgical clipping experienced longer procedure times as compared with those treated with endovascular coiling; however, a previous study reported that thromboembolism occurred more frequently in patients who had undergone endovascular coiling procedures of longer durations [35]. Therefore, we performed a path analysis to determine whether the procedure time itself affected the cognitive function or whether it appeared to be influenced by other factors. It was found that microsurgical clipping was associated with a poor visuospatial function, mediated through prolonged procedure times. Furthermore, the procedure time had a direct effect on the frontal/executive functions. Prolonged procedure times were also associated with a prolonged anaesthesia duration; however, the anaesthesia duration did not affect the cognitive functions. Therefore, through our study, it was found that prolonged procedure times had a considerable effect on the cognitive function. Generally, the procedure duration increases the incidence and severity of post-operative cognitive dysfunction [36]. The release of endotoxin stimulates the release of interleukins, which are responsible for a systemic inflammatory response [37]. Immune response to endotoxin was reportedly associated with post-operative cognitive dysfunction [38]. Disruption of cerebral autoregulation might also occur during surgery [39,40]. In particular, hypercapnia, anaemia, and hypothermia all affect autoregulation and can be exacerbated by a prolonged surgery [41,42]. Furthermore, previous studies reported that in the case of surgery for cardiac disease, a prolonged surgical duration was associated with an impaired cognitive function [43,44]. In these studies, researchers assumed that prolonged surgical durations would lead to greater microvascular obstruction, resulting in cognitive impairment.

In this study, a history of hypertension was also associated with poor memory in patients treated for UIAs. Hypertension induces vascular structural alterations, reduces the cerebral blood flow, and exacerbates amyloid deposition (a characteristic pathological finding in Alzheimer’s disease) [45]. Therefore, the relationship between hypertension and cognitive impairment is well known [46]; however, there is still insufficient evidence on the relationship between post-procedural cognitive impairment and hypertension. Although some studies have reported that post-operative delirium [47] (a risk factor for post-operative cognitive impairment) is associated with hypertension, the direct relationship between post-operative cognitive impairment and hypertension remains unclear [48]; however, in this study, a history of hypertension was associated with a lower visual memory score. Therefore, according to the relationship between hypertension and cognitive impairment, it is likely that the brains of patients with UIA and hypertension were relatively more vulnerable to injury. As a result, we assumed that after the treatment, these patients may have a lower cognitive function as compared with those without hypertension.

We evaluated the cognitive function in the acute stage of treatment to determine the effects on the post-treatment acute stage. Most of the existing studies have focused on identifying the long-term prognosis by conducting cognitive function assessments after 3 months or 1 year [16,17]; however, we attempted to investigate the acute effects before these time points. The cognitive deficit caused by brain injury, and thus, the symptoms of patients with brain injury can change in nature over time due to the plasticity of the brain [14]. Owing to a rapid recovery of the cognitive function, we thought that it would be more useful to evaluate the acute stage to confirm the effect of risk factors on the domain-specific cognitive function. In the case of language function, a study reported that dysfunction appeared only 1–4 weeks after aneurysm rupture [49]. Furthermore, since the degree of cognitive function in the acute phase is correlated with the cognitive function after 6 months [12], it would be clinically meaningful to treat people exhibiting risk factors for cognitive impairment in the acute phase.

This study has several limitations. First, since it had a small sample size, we may have missed some significant relationships. In particular, due to the small sample size, we could not analyse the effects of aneurysms on the cognitive function according to the aneurysm location. Furthermore, the findings in patients with UIA are subject to careful interpretation and generalisation, since the sample size was very small. Second, there was no adequate matching of patients with SAH and UIA. Therefore, we could not directly compare the cognitive functions of patients with SAH and UIA; however, we divided these factors into groups and explored only the ones affecting each group. Third, although the patients who underwent microsurgical clipping took a longer period of recovery than those who underwent endovascular coiling, they were generalised in the treated group; however, to receive the detailed neuropsychological evaluation conducted in our study, it is difficult to say that the recovery of the general condition of the patients has not improved since it can be received in a state when they can sit and concentrate for about 1.5 to 2 h. Moreover, there was no difference in the MMSE score, a general cognitive assessment, between the group that received the surgical treatment and the group that received the endovascular treatment (surgical treatment 25.3, endovascular treatment 23.8, *p* value = 0.255). Fourth, in this study, a pre-treatment neuropsychological test was not performed, making it impossible to compare the pre-treatment and post-treatment cognitive functions. The best study outcome is an evaluation of the post-treatment cognitive impairment through a comparison of the pre-treatment and post-treatment results; however, in the case of SAH, the ruptured intracranial aneurysm occurs suddenly, and it was almost impossible to conduct a pre-treatment assessment. In the case of UIA, the pre-treatment assessment may be possible, and we will consider this in a future study. Fifth, we could not evaluate the detailed microstructural changes, since only CT and simple magnetic resonance imaging were used for post-treatment imaging. Finally, since this study was cross-sectional in nature, additional studies are needed on the long-term effects of all discussed factors on the cognitive function. Since SAH is a dynamic state which is affected by numerous factors, cognitive function should be measured over a longer period in a future study.

Despite these limitations, since we assessed the cognitive function in the acute stage of treatment for intracranial aneurysms, we were able to note that the visuospatial, memory, and frontal/executive functions were impaired in patients treated for intracranial aneurysms. Among the clinical factors, a history of hypertension, the dome-to-neck ratio, aneurysms located in the posterior circulation, treatment with microsurgical clipping, procedure times, anaesthesia durations, vasospasms, and new infarctions were observed to affect the cognitive function. Therefore, to reduce cognitive deficits after the intracranial aneurysm treatment, more consideration should be given to procedural method selection, procedure time shortening, and complication prevention.

## 5. Conclusions

In the acute stage of treatment, patients treated for intracranial aneurysms showed a greater cognitive impairment in terms of the visuospatial function, verbal memory, visual memory, and frontal/executive domains. Among the clinical factors, a history of hypertension, the dome-to-neck ratio, aneurysms located in the posterior circulation, treatment with microsurgical clipping, procedure times, anaesthesia duration, vasospasms, and new infarctions were associated with cognitive function decline.

## Figures and Tables

**Table 1 jcm-11-05053-t001:** Baseline demographic and clinical characteristics.

Characteristics	UIA (*n* = 14)	SAH (*n* = 34)
Age (years) ^a^	65.5 ± 8.79	56.6 ± 7.74
Gender (% ^b^, female)	11 (78.6)	25 (73.5)
Years of education (years) ^a^	7.00 ± 4.038	9.22 ± 5.119
PHASES score ^a^	7.2 ± 1.81	8.4 ± 1.91
WFNS grade (%)		
1		14 (41.2)
2		14 (41.2)
3		5 (14.7)
4		1 (2.9)
Vascular risk factor ^b^		
Hypertension (%)	10 (71.4)	11 (32.4)
Diabetes mellitus (%)	2 (14.3)	0 (0.0)
Dyslipidemia (%)	3 (21.4)	2 (5.9)
Procedural time (min) ^a^	120.4 ± 54.30	122.7 ± 101.31
Anaesthetic duration (min) ^a^	173.2 ± 61.04	166.9 ± 125.51
Interval between disease onset and NP test (days) ^a^	21.7 ± 13.08	23.3 ± 13.76
Microsurgical clipping (%)	11 (78.6)	14 (41.2)
Complications ^b^		
Hydrocephalus	0 (0.0)	5 (14.7)
Vasospasm	0 (0.0)	4 (11.8)
New infarction	3 (21.4)	19 (55.9)
Delayed cerebral ischemia	0 (0)	0 (0)

^a^ Values are presented as mean ± standard deviation, ^b^ Number of cases with percentages in parentheses. WFNS: World Federation of Neurological Surgeons; UIA: Unruptured intracranial aneurysm; SAH: Subarachnoid haemorrhage; NP test: Neuropsychological test.

**Table 2 jcm-11-05053-t002:** Location of the intracranial aneurysms.

	UIA	SAH
Single aneurysm	12 (85.7%)	30 (88.2%)
ICA	2	2
AcomA	2	12
ACA	0	1
MCA	5	6
PCA	0	0
PcomA	3	7
PICA	0	1
VA	0	1
Multiple aneurysms	2 (14.3%)	4 (11.8%)
	MCA + Acom	MCA + PcomA
	ICA + MCA	AcomA + ICA
		MCA + MCA
		PCA + ICA

ICA: Internal carotid artery; AcomA: Anterior communicating artery; ACA: Anterior cerebral artery; MCA: Middle cerebral artery; PCA: Posterior cerebral artery; PcomA: Posterior communicating artery; PICA: Posterior inferior cerebellar artery; VA: Vertebral artery.

**Table 3 jcm-11-05053-t003:** Neuropsychological test scores for patients with intracranial aneurysm.

	UIA (*n* = 14)		SAH (*n* = 34)	
	Mean Raw Score (SD)	Mean z Score (SD)	Mean Raw Score (SD)	Mean z Score (SD)
Digit span forward	4.9 ± 1.77	−0.44 ± 1.189	5.6 ± 1.65	−0.41 ± 1.020
Digit span backward	3.1 ± 1.70	−0.46 ± 1.505	3.9 ± 1.59	−0.50 ± 1.651
K-BNT	41.6 ± 11.77	−0.42 ± 1.252	46.5 ± 9.55	−0.62 ± 1.472
RCFT copy	27.07 ± 7.516	−0.58 ± 0.929	29.38 ± 6.700	−1.10 ± 1.922
SVLT immediate recall	15.8 ± 5.96	−0.75 ± 1.278	17.0 ± 5.52	−1.12 ± 1.142
SVLT delayed recall	3.7 ± 2.87	−0.99 ± 1.147	4.0 ± 3.10	−1.45 ± 1.227
SVLT recognition	19.9 ± 3.08	−0.28 ± 1.593	19.7 ± 2.66	−0.99 ± 1.322
RCFT immediate recall	6.71 ± 6.275	−1.02 ± 1.025	13.58 ± 7.633	−0.56 ± 1.044
RCFT delayed recall	7.21 ± 6.284	−0.97 ± 1.057	12.61 ± 7.369	−0.73 ± 1.033
RCFT recognition	18.4 ± 2.31	−0.51 ± 1.057	19.0 ± 2.50	−0.64 ± 1.321
COWAT animal	9.6 ± 4.96	−1.29 ± 1.176	11.7 ± 5.31	−1.18 ± 1.098
COWAT supermarket	12.9 ± 6.74	−0.58 ± 1.277	14.5 ± 6.48	−0.81 ± 0.960
COWAT phonemic	14.7 ± 10.17	−0.91 ± 1.196	16.8 ± 10.05	−1.23 ± 0.875
Stroop test colour reading correct	51.7 ± 34.12	−1.60 ± 1.833	65.0 ± 34.12	−1.83 ± 2.050

Values are presented as mean ± standard deviation. UIA: Unruptured intracranial aneurysm; SAH: Subarachnoid haemorrhage; K-BNT: Korean version of the Boston Naming Test; RCFT: Rey-Osterrieth Complex Figure Test; SVLT: Seoul Verbal Learning Test; COWAT: Controlled Oral Word Association Test.

**Table 4 jcm-11-05053-t004:** Analysis estimating factors associated with cognitive impairment in patients with UIA.

	RCFT Immediate Recall		COWAT Animal		Stroop Test	
	β (SE)	*p* Value	β (SE)	*p* Value	β (SE)	*p* Value
Size of aneurysm	0.565 (1.155)	0.635	1.279 (0.776)	0.130	7.997 (5.656)	0.195
Dome-to-neck ratio of aneurysm	0.094 (0.330)	0.788	−0.004 (0.249)	0.987	2.486 (1.591)	0.162
Located in the post. circulation	N/A		N/A		N/A	
Interval between disease onset and NP test	−0.014 (0.146)	0.926	−0.001 (0.109)	0.996	−0.118 (0.923)	0.901
Microsurgical clipping	−0.969 (4.476)	0.833	1.021 (3.344)	0.766	−15.171 (24.448)	0.552
Procedure time	−0.026 (0.033)	0.453	−0.017 (0.025)	0.522	−0.244 (0.164)	0.175
Anaesthetic duration	−0.011 (0.031)	0.720	−0.001 (0.023)	0.957	−0.148 (0.160)	0.381
Multiple procedure	−8.634 (6.383)	0.206	−0.491 (5.197)	0.927	−22.365 (36.062)	0.552
Hypertension	−8.235 (3.682)	0.049	−5.314 (2.930)	0.100	−45.909 (23.146)	0.083
Diabetes mellitus	0.283 (5.408)	0.959	−1.384 (4.027)	0.738	−13.277 (33.967)	0.706
Dyslipidaemia	−0.724 (4.762)	0.882	1.741 (3.528)	0.632	−10.077 (30.106)	0.746
Hydrocephalus	N/A		N/A		N/A	
Vasospasm	N/A		N/A		N/A	
New infarction	1.832 (4.398)	0.686	−2.021 (3.260)	0.549	−18.186 (28.516)	0.541

Linear regression analysis was adjusted by age and years of education. Each analysis was performed with a single model including age, years of education, and each factor. UIA: Unruptured intracranial aneurysm; RCFT: Rey-Osterrieth Complex Figure Test; COWAT: Controlled Oral Word Association Test; post.: Posterior; NP: Neuropsychological; N/A: Not applicable.

**Table 5 jcm-11-05053-t005:** Analysis estimating factors associated with cognitive impairment in patients with SAH.

	RCFT Copy		SVLT Immediate Recall		SVLT Delayed Recall		COWAT Animal		COWAT Phonemic		Stroop Test	
	β (SE)	*p* Value	β (SE)	*p* Value	β (SE)	*p* Value	β (SE)	*p* Value	β (SE)	*p* Value	β (SE)	*p* Value
Size of aneurysm	0.110 (0.431)	0.800	0.174 (0.386)	0.656	−0.218 (0.221)	0.331	0.326 (0.325)	0.324	−0.085 (0.688)	0.902	−1.375 (2.206)	0.538
Dome-to-neck ratio of aneurysm	0.671 (0.987)	0.502	0.852 (0.895)	0.350	0.094 (0.518)	0.857	2.109 (0.639)	0.002	1.033 (1.510)	0.501	3.697 (4.924)	0.460
Located in the post. circulation	−5.450 (3.391)	0.119	−3.458 (3.124)	0.277	−3.433 (1.737)	0.057	−2.753 (2.671)	0.311	−8.667 (5.055)	0.099	−39.191 (16.207)	0.022
Interval between disease onset and NP test	−0.051 (0.080)	0.531	0.006 (0.071)	0.928	−0.053 (0.040)	0.196	−0.053 (0.060)	0.385	−0.071 (0.132)	0.592	−0.390 (0.389)	0.325
Microsurgical clipping	−4.192 (1.892)	0.035	−1.122 (1.781)	0.533	−0.517 (1.034)	0.621	−4.074 (1.335)	0.005	−5.619 (3.024)	0.075	−17.471 (9.784)	0.085
Procedure time	−0.030 (0.008)	0.001	−0.008 (0.009)	0.363	−0.006 (0.005)	0.276	−0.009 (0.007)	0.226	−0.035 (0.014)	0.020	−0.090 (0.049)	0.075
Anaesthetic duration	−0.024 (0.007)	0.001	−0.005 (0.007)	0.521	−0.004 (0.004)	0.303	−0.006 (0.006)	0.306	−0.026 (0.012)	0.034	−0.066 (0.040)	0.108
Multiple procedure	−4.132 (2.470)	0.105	0.336 (2.328)	0.886	−0.333 (1.348)	0.807	0.287 (1.985)	0.886	−4.583 (4.137)	0.279	−8.910 (14.046)	0.531
Hypertension	−1.466 (2.248)	0.519	−0.901 (1.997)	0.655	−1.334 (1.135)	0.249	0.847 (1.702)	0.623	0.985 (3.587)	0.786	−11.808 (11.180)	0.300
Diabetes mellitus	N/A		N/A		N/A		N/A		N/A		N/A	
Dyslipidemia	1.110 (4.246)	0.796	−1.422 (3.809)	0.712	−1.950 (2.184)	0.379	0.897 (3.252)	0.785	−6.816 (6.221)	0.284	−24.677 (20.855)	0.247
Hydrocephalus	−0.315 (3.070)	0.919	−5.123 (2.389)	0.040	−3.237 (1.364)	0.024	-−3.112 (2.113)	0.151	−5.320 (5.092)	0.306	−25.910 (14.889)	0.093
Vasospasm	−4.304 (2.980)	0.159	1.779 (2.730)	0.520	0.971 (1.583)	0.544	6.226 (2.051)	0.005	1.559 (5.502)	0.779	6.745 (17.617)	0.705
New infarction	2.725 (1.919)	0.166	2.168 (1.713)	0.215	0.585 (1.013)	0.568	3.746 (1.334)	0.009	3.923 (3.040)	0.209	14.334 (9.759)	0.153

Linear regression analysis was adjusted by age and years of education. Each analysis was performed with a single model including age, years of education, and each factor. SAH: Subarachnoid haemorrhage; RCFT: Rey-Osterrieth Complex Figure Test; SVLT: Seoul Verbal Learning Test; COWAT: Controlled Oral Word Association Test; post.: Posterior; NP: Neuropsychological; N/A: Not applicable.

## Data Availability

The dataset used and analysed during the current study is available from the corresponding author on reasonable request.

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
