# Peer review of "Clinical Factors Contributing to Cognitive Function in the Acute Stage after Treatment of Intracranial Aneurysms: A Cross-Sectional Study"

_jcm, 2022, doi:10.3390/jcm11175053_

Round 1
Reviewer 1 Report
This is a retrospective study that asks the question whether patients with aneurysms of the brain and patients with aneurysmal SAH have cognitive impairment noted after the intervention/illness. Despite that the study is retrospective, it appears the hospital system dose systematic cognitive assessments that most hospitals do not do making this data valuable. There are a number of issues that I feel the authors need to address. I will list them as major and minor below.
Major:
1) In the introduction, the authors make the claim that SAH is a fatal disease, that 51% of patients die and that 80% have cognitive decline. After reviewing the citations, I believe those statements should be revised. SAH is not routinely fatal, the mortality is much lower over the last 15 years and the paper cited about cognitive decline shows that 93% have cognitive impairment. These facts make the argument for the rest of the paper and should be corrected.
2) The authors use the term cognitive decline but are in fact showing data for cognitive impairment. In order to show decline, they would have to have done neuropsychological evaluation prior to aneurysm intervention or SAH.
3) The authors claim in the abstract that "All patients underwent neuropsychological tests 3 months after treatment", a claim that is repeated in the discussion. In Table 1, the interval between disase onset and NP test (days) is 21.7 +/- 13.08 for UIA and 23.3 +/- 13.76 for SAH. This would suggest that few values were in the 2-3 month range. This is a major point of the paper since some patients are still quite ill with SAH in the first 3 weeks (21 days) which could skew this data. A better description of when the tests were performed and how far after hospital discharge would be informative.
Minor:
1) Figure 5 did not reproduce correctly making it hard to read.
2) It would be nice to find a graphical way to represent the data in Figures 3, 4, and 5. The table format is difficult to look at. If the table is preferred, highlighting significant data would be helpful for the reader. One suggestion would be to graphically represent the data and put the tables in a supplemental section.
Reviewer 2 Report
The authors presented a manuscript about the impact of clinical factors on cognitive function in patients who underwent intracranial aneurysm treatment.
The main limitations of this manuscript are:
- small number of patients (48 patients),
- there is no adequate matching of patients with ruptured and unruptured IAs,
- patients were generalized in treated group, although there is an enormous difference in period of recovery after surgical vs endovascular treatment, patients which underwent surgery need longer period of recovery,
- absence of pretreatment neurophysiological testing,
- absence of pretreatment diagnostics (CT, MR)
- SAH is a dynamic state which is affected by numerous factors, so that the patient recovery and cognitive function should be measured not only after 3 months but over a longer period.
I advise the authors to try to implement these suggested changes and to submit the revised manuscript.
Round 2
Reviewer 2 Report
The authors have resolved my concerns and implemented the clarifications throughout the manuscript.